# M³ViT: Mixture-of-Experts Vision Transformer for Efficient Multi-task Learning with Model-Accelerator Co-design

**Hanxue Liang**[1]\*, **Zhiwen Fan**[1]\*, **Rishov Sarkar**[2], **Ziyu Jiang**[3], **Tianlong Chen**[1],
**Kai Zou**[4], **Yu Cheng**[5], **Cong Hao**[2], **Zhangyang Wang**[1]

[1]University of Texas at Austin, [2]Georgia Institute of Technology
[3]Texas A&M University, [4]Protagolabs Inc, [5]Microsoft Research
{haliang,zhiwenfan,tianlong.chen,atlaswang}@utexas.edu
{rishov.sarkar,callie.hao}@gatech.edu, jiangziyu@tamu.edu
kz@protagolabs.com, yu.cheng@microsoft.com

## Abstract

Multi-task learning (MTL) encapsulates multiple learned tasks in a single model and often lets those tasks learn better jointly. However, when deploying MTL onto those real-world systems that are often resource-constrained or latency-sensitive, two prominent challenges arise: (i) during **training**, simultaneously optimizing all tasks is often difficult due to gradient conflicts across tasks, and the challenge is amplified when a growing number of tasks have to be squeezed into one compact model; (ii) at **inference**, current MTL regimes have to activate nearly the entire model even to just execute a single task. Yet most real systems demand only one or two tasks at each moment, and switch between tasks as needed: therefore such "all tasks activated" inference is also highly inefficient and non-scalable.

In this paper, we present a model-accelerator **co-design** framework to enable efficient on-device MTL, that tackles **both** training and inference bottlenecks. Our framework, dubbed **M³ViT**, customizes mixture-of-experts (MoE) layers into a vision transformer (ViT) backbone for MTL, and sparsely activates task-specific experts during training, which effectively disentangles the parameter spaces to avoid different tasks' training conflicts. Then at inference with any task of interest, the same design allows for activating only the task-corresponding sparse "expert" pathway, instead of the full model. Our new model design is further enhanced by hardware-level innovations, in particular, a novel computation reordering scheme tailored for memory-constrained MTL that achieves zero-overhead switching between tasks and can scale to any number of experts. Extensive experiments on PASCAL-Context [1] and NYUD-v2 [2] datasets at both software and hardware levels are conducted to demonstrate the effectiveness of the proposed design. When executing single-task inference, M³ViT achieves higher accuracies than encoder-focused MTL methods, while significantly reducing **88%** inference FLOPs. When implemented on a hardware platform of one Xilinx ZCU104 FPGA, our co-design framework reduces the memory requirement by **2.40×**, while achieving energy efficiency up to **9.23×** higher than a comparable FPGA baseline. Code is available at: https://github.com/VITA-Group/M3ViT.

## 1 Introduction

Vision Transformers (ViTs) [3, 4, 5, 6], as the latest performant deep models, have achieved impressive performance on various computer vision tasks [7, 8, 9]. These models are specially trained or tested

---

\*Equal contribution

for only one or a few tasks; however, many real-world applications require one compact system that can *handle many different tasks* efficiently, and often need to swiftly *switch between tasks* per demand. For example, an autonomous driving system [10] needs to perform and switch between many tasks such as drivable area estimation, lane detection, pedestrian detection, and scene classification: apparently both single task inference and cross-task switching need to happen at ultra-low latency. As another example, smart-home indoor robots [11] are expected to address semantic segmentation, navigation, tracking, or other tasks in varying contexts, with very limited on-board resources. Multi-task learning (MTL) [12, 13, 14] solves multiple tasks simultaneously within a single model and learns improved feature representations [15] shared by related tasks [16, 17]. Therefore, accomplishing **realistic efficient MTL** is becoming a key knob for building real-time sophisticated AI systems.

Despite the promise, challenges persist to build an efficient MTL model suitable for real-world applications: ❶ during **training**, prior works [18, 19, 20] indicate the competition of different tasks in training may degrade MTL, since the same weights might receive and be confused by conflicting update directions. Specifically, [19] reveals that negative cosine similarities between different tasks' gradients are detrimental. [21, 22] confirm that conflicting gradients not only slow down convergence but also bias the learned representations against some tasks. That is only getting worse on compact models owing to their limited modeling capacity. To tackle the cross-task conflicts, solutions have been proposed by varying learning rate speeds of different tasks [20], using "cross-stitch" sharing [23], or re-balancing task gradients [19, 24, 20, 25]. However, they either require task-specific design or significantly increase the model complexity which contradicts our efficiency goal. ❷ at **inference**, existing MTL regimes typically activate the entire backbone model unconditionally. However, **many real systems only need to call upon one or a few tasks at each moment**, hence the "all activated" inference is heavily inefficient and non-scalable. For example, current regimes [14, 23, 26, 27] have to activate the whole gigantic ResNet [28] encoder even just to execute a single monocular depth estimation task or so. If the number of tasks scale up [29] and the backbone keeps growing bigger, the "per task" inference efficiency of the resultant MTL model could become catastrophically poor.

To tackle these bottlenecks, we propose a model-accelerator **co-design** framework that enables efficient on-device MTL. Specifically, in the software level, we propose to adapt mixture of experts (MoE) layers [30, 31] into the MTL backbone, as MoE can adaptively divide-and-conquer the entire model capacity into smaller sub-models [30, 32]. Here, we replace the dense feed-forward network in the ViT with sparsely activated MoE experts (MLPs). A task-dependent gating network will be trained to select the subset of experts for each input token, conditioning on tasks. During training, this task-dependent routing principle effectively disentangles the parameter spaces, balancing feature reuse and automatically avoiding different tasks' training conflicts. Meanwhile, at the inference stage with any task of interest, this design naturally allows for sparse activation of only the experts corresponding to the task instead of the full model, thus achieving highly sparse and efficient inference for the specific task. In the hardware level, we propose a novel computation reordering mechanism tailored for memory-constrained MTL and MoE, which allows scaling up to any number of experts and also achieves zero-overhead switching between tasks. Specifically, based on ViT, we push tokens to per-expert queues to enable expert-by-expert computation rather than token-by-token. We then implement a double-buffered computation strategy that hides the memory access latency required to load each expert's weights from off-chip memory, regardless of task-specific expert selection. This design naturally incurs no overhead for switching between frames or tasks in FPGA.

To validate the effectiveness, we evaluate our performance gain using the ViT-small backbone on the NYUD-v2 and PASCAL-Context datasets. On the NYUD-v2 dataset with two tasks, our model achieves comparable results with encoder-focused MTL methods while reducing 71% FLOPs for single-task execution. When we evaluate on the PASCAL-Context dataset with more tasks, our model achieves even better performance (2.71 *vs.* 0.60) and reduces 88% inference FLOPs. We found the MTL performance gain brought by MoE layers consistently increases as the task count grows. When implemented on a hardware platform of one Xilinx ZCU104 FPGA, our co-design framework reduces the memory requirement by 2.40× while achieving energy efficiency (as the product of latency and power) up to 9.23× higher than comparable FPGA baselines and up to 10.79× higher than the GPU implementation. Our contributions are outlined below:

- We target the problem of efficient MTL, and adopt the more realistic inference setting (activating one task at a time, while switching between tasks). We introduce MoE as the unified tool to attain two goals: both resolving cross-task training conflicts (*better MTL performance*), and sparsely activating paths for single-task inference (*better efficiency*).

Specifically for MTL, the MoE layer is accompanied with a *task-dependent* gating network to make expert selections conditioning on the current task.

- We implement the proposed MTL MoE ViT framework on a hardware platform of one Xilinx ZCU104 FPGA, which enables us to exploit a memory-efficient computation reordering scheme that consolidates per-expert Multiply-and-ACcumulate (MAC) operations such that only one expert's weights are needed on-chip at a time. Our design is scalable to any number of experts while requiring no frame-switching or task-switching overhead.

- We conduct extensive experiments to justify its inference effectiveness in both accuracy and on-edge efficiency metrics. Our framework, dubbed $M^3$ViT, achieves higher accuracies than encoder-focused MTL methods, while significantly reducing **88%** inference FLOPs; on hardware, it reduces the memory requirement by **2.40×** and costs up to **9.23×** and **10.79×** less energy compared to the FPGA and GPU baselines, respectively.

## 2 Related Works

**Multi-task Learning** The generic multi-task learning problem has been studied for a long history. Some non-deep learning-based methods propose to use distance metric [33, 34, 35], probabilistic prior [36, 37, 38, 39, 40] to model the common information among tasks. With the emergence of the deep learning technique, MTL [14, 23, 41, 42, 43, 44] is performed to learn shared representation among tasks. The emergence of ViT further makes it possible to extend the task range from only vision tasks to other modalities tasks (e.g., text, audio) [45, 46, 47, 48, 49]. Current MTL models can be roughly categorized into two types based on where the task interactions take place in the network. The *encoder-focused* architectures [23, 41, 26, 27] only share information in the encoder, before decoding each task with an independent task-specific head. Cross-stitch networks [23] introduce linear combination in each layer. NDDR-CNN [26] improves it by dimensional reduction. MTAN [27] leverages an attention mechanism to learn between tasks. TAPS [50] adapts a base model to a new task by modifying a small task-specific subset of layers. The second type, *decoder-focused* models [43, 44, 51, 52], make initial task predictions in decoder and then leverage features from these initial predictions to further improve output. Although they report higher performance, their models consume a large number of FLOPs, according to [14]. This makes it difficult to deploy them onto those real-world systems that are often resource-constrained or latency-sensitive. And they need to execute all the tasks for initial prediction, which is heavily inefficient in the common scenario when only one or few tasks are needed. Hence, we focus on encoder-focused architecture in this work. Many methods [25, 20, 53, 27] are also proposed to handle the MTL training conflicts problem.

**Mixture of Experts (MoE)** MoE contains a series of sub-models (i.e., experts) and performs conditional computation in an input-dependent fashion [54, 55, 56, 57, 58], based on learned or deterministic routing policies [59, 58]. The traditional dense MoEs suffer from intensive computational costs since they select all experts [60]. Recent studies [30, 61, 62] in natural language processing (NLP) propose sparse MoE that sparsely activates a few experts during both training and inference, thus substantially reducing the cost and allowing gigantic language models even with trillions of parameters [62]. Unfortunately, such a sparse-gated manner still has limitations of unstable training and imbalanced selections among experts. Various solutions are invented from regularization [63, 61, 62] and optimization [64, 65] perspectives. Moreover, MoE has drawn increasing popularity in computer vision [60, 66, 67, 68, 69, 70, 71], where it mainly focuses on considerably smaller network backbones compared to the ones in NLP. For instance, [68] and [69] formulate the channel and kernel of convolutional layers as experts and establish the MoE framework. Several pioneer investigations also explore MoE for multi-task learning, which are related to this work. Particularly, [17, 72, 73] introduce task-specific gating networks to choose different parts of models for processing information from each task. They present certain possibilities of using MoE to solve MTL problems in some cases like classification for medical signals [72], digital number images (MNIST) [73], and recommendation systems [17]. We make a further attempt to adapt MoE into a compact model for dense prediction multi-task learning, along with software-hardware co-design.

**Vision Transformer** There are growing interests in exploring the use of transformers [74, 3] for computer vision tasks since its success in the natural language processing [74, 75, 76], including image generation [77, 78], generative adversarial networks [79, 80], image classification [77, 3, 81, 82, 83, 84, 82, 85], semantic segmentation [8, 86], object detection [6, 87], 3D data processing [88, 89, 90], novel view synthesis [91, 92], and many others [93, 94, 95, 96].

**Hardware**    FPGA acceleration of Transformer-based models has attracted increasing attention. Pioneering works [97, 98, 99, 100] note that transformers are computation- and memory-intensive and are too large to fit on the FPGA. Therefore, various model compression methods have been proposed, such as activation quantization, token pruning, block-circulant matrices (BCM) for weights, block-balanced weight pruning, and column-balanced block weight pruning. Such compression methods are lossy and require compression-aware training to regain accuracy. To our best knowledge, there is *no existing FPGA accelerator for MoE* in a Transformer-based model. The MoE mechanism exposes great challenges to FPGA since it requires swift expert switching between tokens and frames, which may introduce significant overhead of memory and parameter loading. In this work, however, we propose a novel expert-by-expert computation-reordering approach that can reduce the overhead to negligible despite the number of experts, and does not require model compression or re-training.

## 3    Method

**Overview**    We first describe the standard Vision Transformer and MoEs, and then show the proposed MoE ViT design for MTL. To enable dynamically adapting between different tasks with minimum overhead on FPGA, we detail the hardware implementation. Figure 1 shows the whole framework.

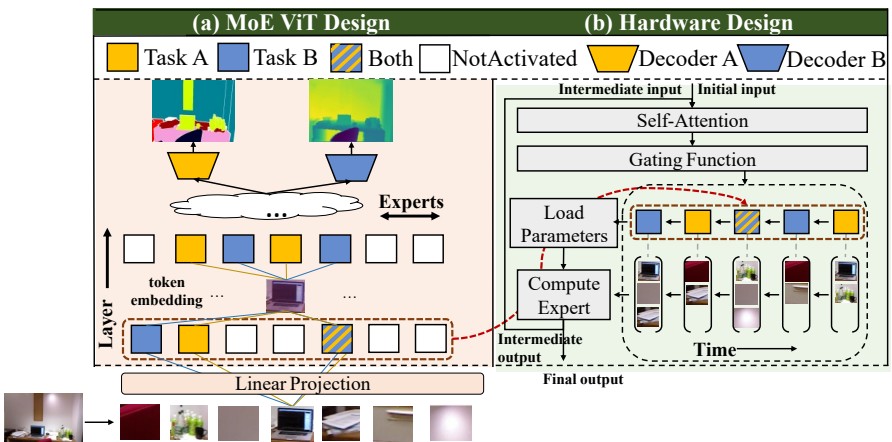

Figure 1: **The overall structure of the proposed M³ViT pipeline**. The input image is split into fixed-size patches, embedded, and combined with position embeddings. In training, the MTL MoE ViT adaptively activates the model by sparsely selecting relevant experts using its task-dependent routers. During inference, only one task will be performed at a time. The hardware collects all patches allocated for each expert and processes them expert-by-expert with the "load parameters" and "compute expert" modules.

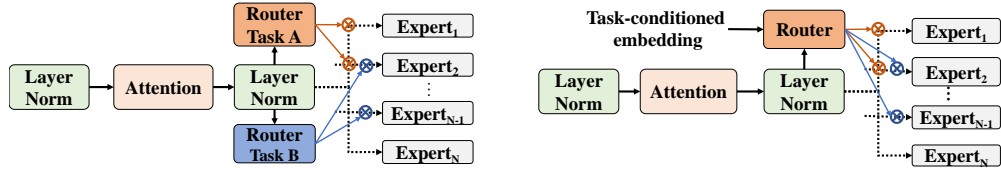

**(a) Multi-gate MoE layer design.**          **(b) Task-conditioned MoE layer design.**

Figure 2: **The proposed two variants of MTL MoE layers**. In the left figure, each task selects its experts using its own router. In the right one, all tasks share one router, while a task-specific embedding is concatenated with the token embedding to formulate the input of the shared router.

### 3.1    Task-dependent MoE ViT Design

**Vision Transformer**    The representative Vision Transformer architecture [3] first splits the input image into non-overlapped patches and projects the patches to a higher hidden dimension using one

convolutional layer. The projected patches (a.k.a. tokens) are then passed through several consecutive transformer layers. Each layer contains a self-attention module and a feed-forward network (MLPs). The self-attention is computed using the scaled-dot product:

$$\text{Attention}(\boldsymbol{Q}, \boldsymbol{K}, \boldsymbol{V}) = \text{softmax}\left(\frac{\boldsymbol{Q}\boldsymbol{K}^T}{\sqrt{C}}\right)\boldsymbol{V} \tag{1}$$

where $\boldsymbol{Q}, \boldsymbol{K}, \boldsymbol{V} \in \mathbb{R}^{N \times C}$ are the query, key and value matrices computed from input tokens; $N$ and $C$ indicate the token number and the hidden dimension. In our experiments, we adopt the DeiT [4] as the backbone encoder, which is a data-efficient ViT variant that distills tokens to ensure the student learns from the teacher through attention.

**Mixture of Experts Layer**   A Mixture of Experts (MoE) layer typically consists a group of $N$ experts $f_1, f_2, \cdots, f_N$ along with a router $\mathcal{R}$ (or gating network) to select the corresponding experts. The experts network stands for multi-layer perceptrons [62, 101] in ViTs. The router $\mathcal{R}$ plays a key role within our MoE ViT design as it determines task routings via only sparsely activating relevant experts. We adopt a representative router called top-$K$ gating [30] based on ViT. With input $\boldsymbol{x}$, the resultant output of MoE layers can be formulated as the summation of the selected top $K$ experts from $N$ expert candidates using a router:

$$y = \sum_{k=1}^{K} \mathcal{R}(\boldsymbol{x})_k \cdot f_k(\boldsymbol{x}), \tag{2}$$

$$\mathcal{R}(\boldsymbol{x}) = \text{TopK}(\text{softmax}(\mathcal{G}(\boldsymbol{x}), K)), \tag{3}$$

$$\text{TopK}(\boldsymbol{v}, K) = \begin{cases} \boldsymbol{v} & \text{if } \boldsymbol{v} \text{ is in the top } K \text{ elements} \\ 0 & \text{otherwise} \end{cases} \tag{4}$$

where $\mathcal{G}$ represents the learnable network within the router, for which we employ a single-layer MLP in practice. The $\text{softmax}(\cdot)$ together with $\text{TopK}(\cdot, K)$ sets all elements of the vector to zero except the elements with the largest $K$ values. In practice, we choose $K = 4$ out of $N = 16$ expert candidates. Each expert is computed with $W_2 \sigma_{\text{gelu}}(W_1 x)$, where $\sigma_{\text{gelu}}$ is the GELU activation [102]. $W_1$ and $W_2$ are two learnable weight matrices. Note that we scale down the expert size by four times compared to that in standard ViT MLP layers to make the computation FLOPs equivalent. We also employ the load and important balancing loss with the weight of 0.01 following [30] to avoid always picking the same experts while ignoring others. This loss term is also employed for the two task-dependent MTL MoE designs that we introduce next.

**Multi-gate MoE ViT for MTL**   MoE brings training dynamics to balance between large capacity and efficiency, by selecting only a subset of experts using the router. To adapt vanilla MoE into our dense prediction MTL framework, we first propose to assign each dense prediction task a router $\mathcal{R}_i$ to specify its own experts, denoted as multi-gate MTL MoE ViT:

$$y_i = \sum_{k=1}^{K} \mathcal{R}^i(\boldsymbol{x})_k \cdot f_k(\boldsymbol{x}) \tag{5}$$

where $i$ denotes task index. Expert candidates $f^k$ are shared across tasks. The flow chart of the multi-gate variant is shown in Figure 2(a); task-dependent routers take as input the shared token embedding and do their expert selections.

**Task-conditioned MoE ViT for MTL**   Conditional encoding has been widely applied to multi-modal [103] and multi-task [104] models. To achieve task-dependent routing with one gating network, we propose the task-conditioned MTL MoE ViT shown in Figure 2(b). Specifically, suppose we have $n$ tasks in training. We manually define a $n$-dimensional one-hot task-conditioned vector. The vector is fed into a two-layer MLP to extract a 64-dimensional task embedding, which is then concatenated with token embeddings to form the task-dependent input for the router in the MoE layer:

$$y_i = \sum_{k=1}^{K} \mathcal{R}(\boldsymbol{x}, \boldsymbol{t_i})_k \cdot f_k(\boldsymbol{x}), \tag{6}$$

$$\boldsymbol{t_i} = \text{ReLU}(\mathcal{T}(\boldsymbol{x}, \boldsymbol{e}_i)) \tag{7}$$

where $\mathcal{T}$ indicates the two-layer MLPs to extract task-conditioned embeddings, $\boldsymbol{e}_i \in \{0, 1\}^n$, and $\sum_{j=1}^{n} e_j = 1$. We denote this conditional design as task-conditioned MTL MoE ViT, in which backbone model parameters do not proportionally increase if we include more tasks in training.

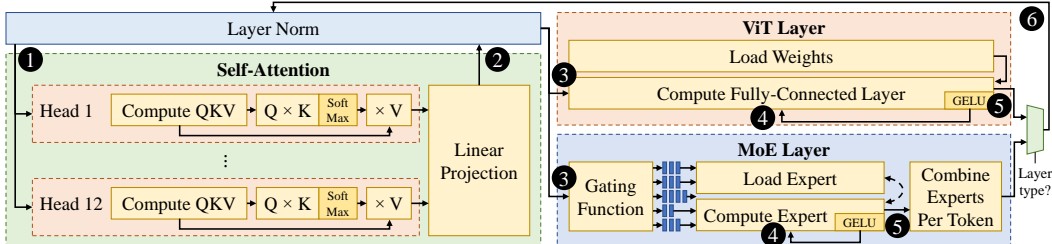

Figure 3: **Hardware implementation of a ViT block of M³ViT.** The hardware implementation consists of a layer norm unit, a self-attention unit containing 12 independent heads followed by a linear projection, a unit to compute the fully-connected layers of a standard ViT layer, and a unit to select and compute the experts in an MoE layer. Numerical indicators within the figure indicate the path through which data flows during the computation of a single layer of M³ViT, either a ViT layer or an MoE layer. This hardware is shared across all blocks.

## 3.2 Circuit-level Implementation

We co-design the hardware to support MTL MoE ViT. We design a layer-wise implementation of M³ViT on a Xilinx ZCU104 FPGA, a diagram of which is shown in Figure 3. The design computes layers sequentially but parallelizes computation steps within each layer. By proposing a novel computation reordering scheme, our hardware design features memory-efficient expert computation that also achieves zero-overhead task switching and frame switching.

**Challenges of Naive Method**   A straightforward (but naive) implementation would compute the output for each token in the order it appears in the input sequence: all tokens choose any $K$ experts out of the $N$ candidates, so ostensibly the only way to avoid data loading overhead would be to keep weights of all $N$ experts on-chip at all times. However, this requires extreme on-chip memory usage, scaling with $O(N)$ and typically exceeding FPGA on-chip memory capacity unless $N$ is very small.

**Challenges of Cache-based Method**   We can adopt a cache to store several experts on-chip at any given time. However, this on-demand approach incurs long delays from off-chip DRAM accesses whenever the cache needs to be repopulated with an expert's weights. Further, we experimentally found that all experts are likely to be activated at least once across all tokens, exhibiting a cache-unfriendly access pattern. Therefore, although a cache-based design alleviates memory inefficiency, it incurs severe delays by frequently loading the weights of experts.

**Proposed Solution: Memory-efficient Computation Reordering**   The crux of the problem lies in the unpredictability of the set of experts that will be needed by tokens at any given time. We address this problem at its root by designing a novel computation reordering scheme that flips the compute pattern on its head: rather than computing the MoE layer token-by-token, we instead compute it expert-by-expert. The overall flow chart of the reordering scheme is shown in Figure 4.

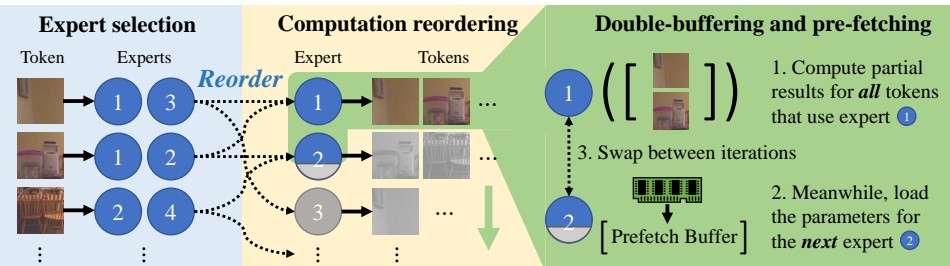

Figure 4: **The computation reordering flow used by M³ViT for hardware memory efficiency.** The MoE gating function selects $K$ experts for each token, which are used to route tokens to per-expert queues. This is followed by a double-buffered computation flow that computes one expert's results on its entire token queue while loading another expert's parameters, swapping buffers between iterations.

Specifically, we propose to add each token to a queue for its selected top-$K$ experts, instead of computing the token output immediately. Our hardware then makes use of the per-expert queues via

Table 1: Comparisons with encoder-focused MTL architectures on the PASCAL-Context dataset.

| Model | Backbone | Seg. (mIoU↑) | Norm. (mErr)↓ | H. Parts (mIoU)↑ | Sal. (mIoU)↑ | Edge (odsF)↑ | $\Delta_m$ (%)↑ | FLOPS (G)↓ | Energy (W·s)↓ |
|---|---|---|---|---|---|---|---|---|---|
| STL-B | ResNet-18 | 66.2 | 13.9 | 59.9 | 66.3 | 68.8 | 0.00 | 167 | 1.029 |
| MTL-B | ResNet-18 | 63.8 | 14.9 | 58.6 | 65.1 | 69.2 | −2.86 | 167 | 1.029 |
| Uncertainty [25] (MTL-B) | ResNet-18 | 65.4 | 16.5 | 59.2 | 65.6 | 68.6 | −4.60 | 167 | 1.029 |
| DWA [53] (MTL-B) | ResNet-18 | 63.4 | 14.9 | 58.9 | 65.1 | 69.1 | −2.94 | 167 | 1.029 |
| GradNorm [20] (MTL-B) | ResNet-18 | 64.7 | 15.4 | 59.0 | 64.5 | 67.0 | −3.97 | 167 | 1.029 |
| MGDA [27] (MTL-B) | ResNet-18 | 64.9 | 15.6 | 57.9 | 62.5 | 61.4 | −6.81 | 167 | 1.029 |
| MTAN [27] | ResNet-18 | 63.7 | 14.8 | 58.9 | 65.4 | 69.6 | −2.39 | 212 | 5.306 |
| Cross-Stitch [23] | ResNet-18 | 66.1 | **13.9** | 60.6 | **66.8** | 69.9 | +0.60 | 647 | 6.001 |
| NDDR-CNN [26] | ResNet-18 | 65.4 | **13.9** | 60.5 | **66.8** | 69.8 | +0.39 | 747 | 5.034 |
| M-ViT (MTL-B) | ViT-small | 70.7 | 15.5 | 58.7 | 64.9 | 68.8 | −1.77 | **83** | 3.062 |
| M²ViT (+MoE) | MoE ViT-small | **72.8** | 14.5 | **62.1** | 66.3 | **71.7** | **+2.71** | 84 | 7.446 |
| M³ViT (+MoE+Codesign) | MoE ViT-small | 72.8 | 14.5 | 62.1 | 66.3 | 71.7 | +2.71 | 84 | **0.690** |

a double-buffered computation flow, also known as ping-pong buffering [105]: one buffer is filled with an expert's weights from off-chip memory accesses, while another already-loaded buffer is used to compute another expert's results for its entire token queue. After both operations finish, the buffers are swapped, and the process repeats.

**Scalability and Efficiency** Our approach hides nearly all latency from off-chip memory accesses to load expert weights, and it uses $O(1)$ on-chip memory with respect to $K$ and $N$, making it scalable to any number of experts. Additionally, our method's efficiency does not rely on any specific usage pattern of experts for a given frame or a given task, so we naturally achieve zero-overhead switches between frames and between tasks. Task switches and frame switches in our hardware design do not change our computation flow at all, and there is no specific step taken to execute the switch.

## 4 Experiments

### 4.1 Experiment Setup

To evaluate the propose method, we conduct experiments on two popular dense labeling MTL benchmarks, i.e. NYUD-v2 [2] and PASCAL-Context [1]. Both datasets are described below.

**Datasets** The **PASCAL-Context** [1] contains a total of 10,103 images, for the five tasks of edge detection (Edge), semantic segmentation (Seg.), human parts segmentation (H.Parts), surface normals (Norm.), and saliency detection (Sal.). The **NYUD-v2 dataset** [2] is an indoor dataset which consists of RGB-D images of 464 indoor scenes. There are 795 images for training and 654 images for testing, both with annotation for semantic segmentation (Seg.) and monocular depth estimation (Depth).

**Evaluation Metrics** For software level evaluation, we adopt the standard evaluation metrics following [14, 106, 51]. Particularly, we use mean intersection over union (mIoU) for semantic segmentation, human parts segmentation, and saliency; mean error (mErr) for surface normals estimation, root mean square error (rmse) for depth estimation; and optimal dataset F-measure (odsF) [107] for edge detection. Following [14], we use $\Delta_m$ to evaluate a MTL model $m$ as the average per task drop with respect to the STL model $b$ over all tasks: $\Delta_m = \frac{1}{T} \sum_i^T (-1)^{l_i} (M_{m,i} - M_{b,i})/M_{b,i}$, where $M_{m,i}$ and $M_{b,i}$ are the metrics of task $i$ for the model $m$ and $b$ respectively, and $l_i = 1$ if a lower value means better performance.

To evaluate our model-accelerator design, we consider latency, energy usage (as the product of latency and power), and on-chip memory usage for single-task inference using a batch size of 1.

**Network Configuration and Implementation Details** We evaluate our model based on several versions of ViT backbone [4] including ViT-tiny, ViT-small, and ViT-base. Our FPGA designs target the Xilinx ZCU104 FPGA at a 300 MHz clock frequency, consuming 10 W of power. The GPU baselines are measured on the NVIDIA Quadro RTX 8000. The results reported below are based on ViT-small. Please refer to the supplementary materials for training setup, more details on network configuration, results on ViT-tiny and ViT-base, and details of our target hardware platforms.

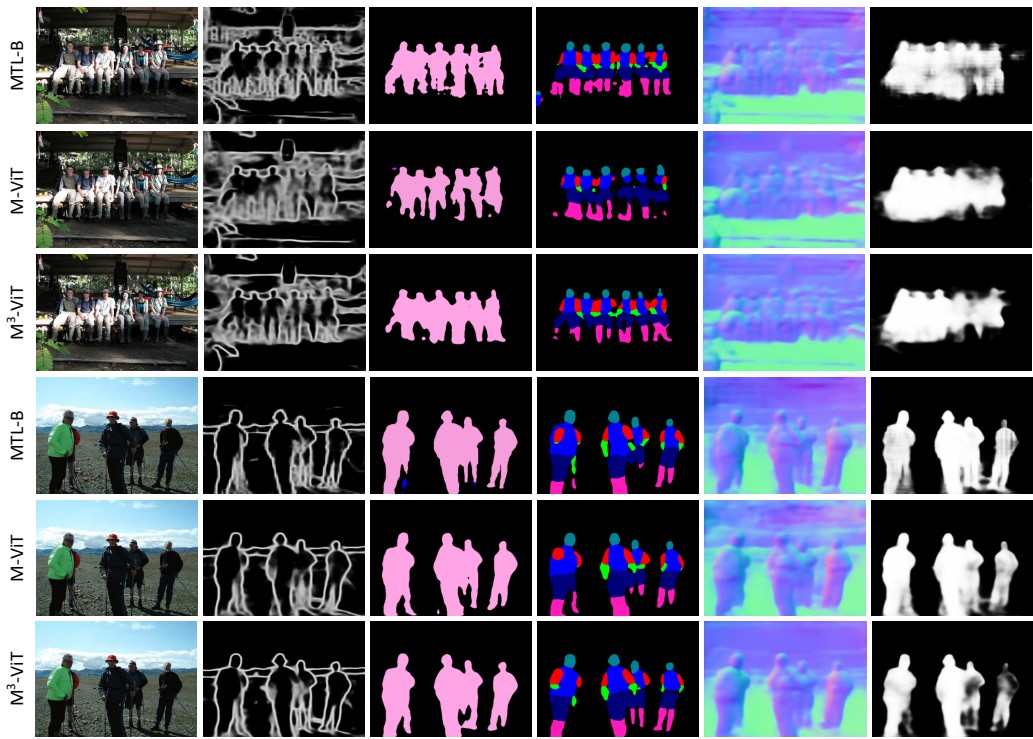

Figure 5: **Qualitative result on PASCAL-Context**. We compare between vanilla MTL-B, M-ViT and $M^2$ViT models, and our model outperforms baseline on edge detection, semantic segmentation and human parts segmentation and saliency detection.

Table 2: Comparisons with encoder-focused MTL architectures on the NYUD-v2 dataset.

| Model | Backbone | Seg. (mIoU)↑ | Depth (rmse)↓ | $\Delta_m$ (%) ↑ | FLOPS (G) ↓ | Energy (W·s) ↓ |
|---|---|---|---|---|---|---|
| STL-B | ResNet-50 | 43.9 | 0.585 | 0.00 | 192 | 2.145 |
| MTL-B | ResNet-50 | 44.4 | 0.587 | +0.41 | 192 | 2.145 |
| Uncertainty [25] (MTL-B) | ResNet-50 | 44.0 | 0.590 | −0.23 | 192 | 2.145 |
| DWA [53] (MTL-B) | ResNet-50 | 44.1 | 0.591 | −0.28 | 192 | 2.145 |
| GradNorm [20] (MTL-B) | ResNet-50 | 44.2 | 0.581 | +1.45 | 192 | 2.145 |
| MGDA [27] (MTL-B) | ResNet-50 | 43.2 | 0.576 | +0.02 | 192 | 2.145 |
| MTAN [27] | ResNet-50 | 45.0 | 0.584 | +1.32 | 320 | 5.036 |
| Cross-Stitch [23] | ResNet-50 | 44.2 | **0.570** | **+1.61** | 310 | 4.221 |
| NDDR-CNN [26] | ResNet-50 | 44.2 | 0.573 | +1.38 | 340 | 4.244 |
| M-ViT (MTL-B) | ViT-small | 40.9 | 0.631 | −6.27 | **100** | 2.097 |
| $M^2$ViT (+MoE) | MoE ViT-small | **45.6** | 0.589 | +1.59 | **100** | 8.189 |
| **$M^3$ViT (+MoE+Co-design)** | MoE ViT-small | 45.6 | 0.589 | +1.59 | **100** | **0.845** |

## 4.2 Comparison with State-of-the-art Dense Prediction MTL

As we target an efficient MTL system under the single-task inference setting, we conduct experiments on encoder-focused architectures (more details in Section 2). MTL-B [14] is a vanilla multi-task learning baseline model which is composed of a shared backbone in combination with task-specific heads. Several state-of-the-art (SoTA) encoder-focused MTL models, including MTAN [27], Cross-Stitch [23] and NDDR-CNN [26], improve MTL-B by proposing feature sharing methods in the encoder. Our methods are all conducted on vanilla MTL-B, namely, applying MTL on ViT (M-ViT), adding task-dependent MoE design ($M^2$ViT), and adding hardware co-design on FPGA ($M^3$ViT). We also compare with previous works that handle the multi-task training conflicts problem, including uncertainty weighting [25], GradNorm [20], DWA [53], and MGDA [27], and they are evaluated on MTL-B. Single task learning baseline (STL-B) is used for MTL performance evaluation $\Delta_m$.

Table 3: Effect of task-dependent MoE design. $M^3$ViT-Single, $M^3$ViT-Multi., and $M^3$ViT-Task-cond. refer to the MTL MoE model with single router, multi routers, and task-conditioned router, respectively.

| PASCAL-Context | Seg. (mIoU↑) | Norm. (mErr)↓ | H. Parts (mIoU)↑ | Sal. (mIoU)↑ | Edge (odsF)↑ | $\Delta_m$ (%)↑ | FLOPS (G)↓ |
|---|---|---|---|---|---|---|---|
| STL-B | 66.2 | **13.9** | 59.9 | **66.3** | 68.8 | 0.00 | 167 |
| M-ViT (MTL-B) | 70.7 | 15.5 | 58.7 | 64.9 | 68.8 | −1.76 | **83** |
| $M^3$ViT-Single | 71.5 | 14.8 | 61.2 | 65.9 | 71.5 | +1.40 | 84 |
| $M^3$ViT-Multi. | **72.8** | 14.5 | **62.1** | **66.3** | 71.7 | **+2.71** | 84 |
| $M^3$ViT-Task-cond. | 72.0 | 14.4 | 61.3 | 65.8 | **71.8** | +2.22 | 85 |

| NYUD-v2 | Seg. (mIoU)↑ | Depth (rmse)↓ | – | – | – | $\Delta_m$ (%)↑ | FLOPS (G)↓ |
|---|---|---|---|---|---|---|---|
| STL-B | 43.9 | **0.585** | – | – | – | 0.00 | 192 |
| M-ViT (MTL-B) | 40.9 | 0.631 | – | – | – | −6.27 | **100** |
| $M^3$ViT-Single | 45.3 | 0.600 | – | – | – | +0.31 | **100** |
| $M^3$ViT-Multi. | **45.6** | 0.589 | – | – | – | **+1.59** | **100** |
| $M^3$ViT-Task-cond. | 45.3 | 0.595 | – | – | – | +0.74 | 101 |

As multi-gate MoE shows better performance than task-conditioned MoE, the reported $M^2$ViT and $M^3$ViT results are based on the multi-gate design.

**Results on PASCAL-Context Dataset**   As shown in Table 1, even using a vanilla MTL-B framework, introducing MoE ($M^2$ViT) can achieve the highest performance over all previous encoder-focused works (+2.71% MTL performance); meanwhile, it significantly reduces their single task inference FLOPs (particularly, reducing Cross-Stitch by 88%). Comparing against Uncertainty [25], DWA [53], GradNorm [20], and MGDA [27], the superior performance of $M^2$ViT demonstrates its strong capacity in handling training conflict. Moreover, leveraging the Model-Accelerator co-design helps us to consume less than one-tenth the energy cost when deploying our model on FPGA. Some qualitative results are shown in Figure 5.

**Results on NYUD-v2 Dataset**   On this dataset, $M^2$ViT can reduce previous SoTA's inference FLOPs by 68% while achieving comparable MTL performance. Introducing MoE to M-ViT helps to enlarge the model capacity without increasing inference FLOPs, which results in a MTL performance boost from −6.27% to +1.59%. With our Model-Accelerator co-design, we also see a nearly tenfold increase in energy efficiency. Results are shown in Table 2.

### 4.3   Effect of Task-dependent MoE Design

To evaluate the effectiveness of our task-dependent MoE design, we compare between several models in Table 3 including STL-B, MTL ViT (M-ViT), $M^3$ViT with one gating function for all the tasks, $M^3$ViT with multi gates, and $M^3$ViT with task-conditioned token input. Results on both PASCAL-Context and NYUD-v2 datasets show that adding MoE layers into ViT with only one gating function for all tasks can already improve the M-ViT model. Making MoE selection task-dependent can further improve the performance, where multi-gating performs better than task-conditioned gating design. Particularly, comparing our model performance with STL-B as well as previous SoTAs in Table 1 and 2, we find that our MoE model better demonstrates its effectiveness when more tasks need to be encapsulated in the system. More results about our model's performance on different numbers of tasks can be found in the supplement.

### 4.4   Hardware Performance Results

Results comparing hardware performance metrics on FPGA and GPU are shown in Table 4. We first discuss our memory efficiency. The naive approach described in Section 3.2 would require 11.610 MiB of on-chip memory (too much for our FPGA platform), but our compute-reordering design achieves the same result using only 4.840 MiB, demonstrating a 2.40× reduction. Moreover, when comparing against a memory-constrained MoE ViT on FPGA without our compute-reordering method (using the cache-based method described in Section 3.2), we see that our method takes 9.23× less latency and energy on the PASCAL-Context dataset and 8.88× less latency and energy on NYUD-v2.

Table 4: Quantitative comparisons of hardware metrics between FPGA and GPU implementations. **CR** indicates usage of our memory-efficient computation reordering method (Section 3.2).

| Platform | Backbone | CR | Memory (MiB) | PASCAL-Context | | NYUD-v2 | |
|---|---|---|---|---|---|---|---|
| | | | | Latency (ms) | Energy (W·s) | Latency (ms) | Energy (W·s) |
| GPU | ResNet-18 | – | 21.336 | 3.489 | 1.029 | – | – |
| GPU | ResNet-50 | – | 44.939 | – | – | 7.270 | 2.145 |
| GPU | ViT-small | – | 42.058 | 10.381 | 3.062 | 7.110 | 2.097 |
| GPU | MoE ViT-small | – | 82.747 | 25.239 | 7.446 | 27.760 | 8.189 |
| FPGA | ViT-small | – | 4.828 | 68.931 | 0.689 | 84.418 | 0.844 |
| FPGA | MoE ViT-small | ✗ | 4.840 | 637.478 | 6.375 | 750.557 | 7.506 |
| FPGA | MoE ViT-small | ✓ | 4.840 | 69.033 | 0.690 | 84.538 | 0.845 |

Our compute-reordering M$^3$ViT on FPGA also beats all GPU baselines in energy efficiency; e.g., it beats MoE ViT on GPU by 10.79× on PASCAL-Context and by 9.69× on NYUD-v2. For discussion on the latency breakdown of M$^3$ViT, please refer to the supplement.

## 5 Conclusion, Discussion of Limitation and Broader Impact

In this paper, we propose a model-accelerator co-design for efficient on-device MTL. By customizing MTL mixture-of-experts layers into a ViT backbone, we sparsely activate task-specific experts in training to mitigate MTL gradient conflicts. For inference, we can activate only the sparse "expert" pathway relevant to the task of interest for efficiency, and can further achieve zero-overhead switching between tasks with our hardware-level co-design. Extensive experiments that show M$^3$ViT surpasses the top-performing encoder-focused MTL methods, reduces 88% FLOPs, and saves more than 8× energy over our baseline. The limitation of our work is that M$^3$ViT is so far mainly evaluated on academic datasets; we will try real applications like autonomous driving in the future. For broader impact, our work can reduce the resource and energy consumption needed for MTL regimes, while still maintaining SOTA performance, which can effectively serve the goal of Green AI.

## Acknowledgment

Zhiwen Fan, Rishov Sarkar, Cong Hao and Zhangyang Wang are in part supported by the DARPA In-Pixel Intelligent Processing (IP2) program.

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
