# Supplementary Materials for M$^3$ViT: Mixture-of-Experts Vision Transformer for Efficient Multi-task Learning with Model-Accelerator Co-design

Hanxue Liang[1,*] Zhiwen Fan[1*], Rishov Sarkar[2], Ziyu Jiang[3], Tianlong Chen[1],
Kai Zou[4], Yu Cheng[5], Cong Hao[2], Zhangyang Wang[1]
[1]University of Texas at Austin, [2]Georgia Institute of Technology
[3]Texas A&M University, [4]Protagolabs Inc, [5]Microsoft Research
`hanxue@gmail.com,{zhiwenfan,tianlong.chen,atlaswang}@utexas.edu`
`{rishov.sarkar,callie.hao}@gatech.edu, jiangziyu@tamu.com`
`kz@protagolabs.com, yu.cheng@microsoft.com`

## A  Implementation Details

### A.1  Scale-up the M$^3$ViT

For the MTL encoder, we evaluate our model based on several variants of ViT following DeiT [1], including ViT-tiny, ViT-small, and ViT-base. The final ViT block's output feature will be fed into decoders for multi-task predictions. We embed MoE expert layers once in every two ViT blocks. The router is a single-layer MLP which maps token embedding to experts' selection probability. In task-conditioned MoE ViT, the task embedding network $\mathcal{T}$ is a two-layer MLP of dimensions 64 and 64. As for MLP decoder, the previous SoTA works [2, 3, 4] uses Deeplab [5] as the decoder for a ResNet backbone. However, Deeplab is defined for Conv backbone and not suitable for ViT encoder output. Therefore, we follow the prior work [6] and use a PUP [6] as decoder, which is a progressive upsampling strategy that alternates conv layers and upsampling operations. Each decoder contains five conv layers (the first four of dimension 256 and the final one of dimension corresponding to task prediction) and four upsampling layers. This decoder is of lighter weight and consumes fewer FLOPs than Deeplab. The output feature of last and second last conv layers will also be used in a multi-tasks feature distillation module. The distillation module will only be used during train stage and deactivated during inference stage, thus adding no extra FLOPs to the whole network.

### A.2  Training Setup

**Pre-training on ImageNet**  During the MTL pre-train stage, all the encoder backbones will be pre-trained on ImageNet and the decoder will be randomly initialized. In the M-ViT models, we use the pre-trained weights provided by DeiT [1] to initialize all the transformer layers and the input linear projection layer in the encoder. In the MoE ViT models, we pre-train our encoder on ImageNet following the same strategy as its counterpart DeiT ViT encoder in [1].

**MTL Training**  For both NYUD-v2 and PASCAL-Context datasets, we adopt a polynomial learning rate decay schedule and employ SGD as the optimizer with initial learning rate 0.002. Momentum and weight decay are set to 0.9 and 0.0001, respectively. The batch size is 16.

---

*Equal contribution

Table 1: Performance of $M^3$ViT on ViT-tiny and ViT-base

| PASCAL-Context | Backbone | Seg. (mIoU↑) | Norm. (mErr)↓ | H. Parts (mIoU)↑ | Sal. (mIoU)↑ | Edge (odsF)↑ | $\Delta_m$ (%)↑ | FLOPS (G)↓ | Energy (W·s)↓ |
|---|---|---|---|---|---|---|---|---|---|
| STL-B | ResNet-18 | 66.2 | **13.9** | 59.9 | 66.3 | 68.8 | 0.00 | 167 | 1.029 |
| MTL-B | ResNet-18 | 63.8 | 14.9 | 58.6 | 65.1 | 69.2 | −2.86 | 167 | 1.029 |
| Cross-Stitch [3] | ResNet-18 | 66.1 | **13.9** | 60.6 | **66.8** | 69.9 | +0.60 | 647 | 6.001 |
| $M^3$ViT | MoE ViT-tiny | 65.3 | 15.2 | 57.9 | 64.2 | 68.5 | −3.53 | **62** | **0.265** |
| $M^3$ViT | MoE ViT-base | **75.2** | 14.8 | **64.5** | 66.1 | **72.6** | **+4.00** | 161 | 2.325 |

| NYUD-v2 | Backbone | Seg. (mIoU)↑ | Depth (rmse)↓ | – | – | – | $\Delta_m$ (%)↑ | FLOPS (G)↓ | Energy (W·s)↓ |
|---|---|---|---|---|---|---|---|---|---|
| STL-B | ResNet-50 | 43.9 | 0.585 | – | – | – | 0.00 | 192 | 2.145 |
| MTL-B | ResNet-50 | 44.4 | 0.587 | – | – | – | +0.41 | 192 | 2.145 |
| TAPS[7] | ResNet-50 | 44.5 | 0.581 | – | – | – | +1.05 | 192 | 2.312 |
| Cross-Stitch [3] | ResNet-50 | 44.2 | 0.570 | – | – | – | +1.61 | 310 | 4.221 |
| $M^3$ViT | MoE ViT-tiny | 40.3 | 0.643 | – | – | – | −9.05 | **74** | **0.351** |
| $M^3$ViT | MoE ViT-base | **49.1** | **0.557** | – | – | – | **+8.32** | 191 | 2.798 |

## A.3   Hardware Details

**Platform Specifications**   Our targeted FPGA, the Xilinx ZCU104 FPGA, has 1,728 DSPs, 504K LUTs, 461K registers, 11 Mbit block RAM, and 27 Mbit UltraRAM. Our GPU used for baseline measurements, the NVIDIA Quadro RTX 8000, has 4,608 CUDA cores and 48 GB of GDDR6 memory. It runs at a clock frequency of 1,395 MHz and consumes 295 W of power.

# B   More Experiment Results

## B.1   Additional Experiments on ViT-tiny and ViT-base

We further evaluate $M^3$ViT on different variants of ViT including ViT-tiny and ViT-base; results are shown in Table 1. We compare against STL-B, MTL-B, and SoTA encoder-focused MTL model TAPS[7], Cross-Stitch [3]. For TAPS, we adopt joint MTL strategy for comparable training longitude. It can be observed that MoE ViT-base increases the SoTA performance by a large margin, achieving +4.00% on PASCAL-Context and +8.32% on NYUD-v2. Meanwhile, it also consumes lower FLOPs compared to previous ResNet-based methods. MoE ViT-tiny consumes much fewer FLOPs than all previous methods (in particular, less than 1/10 FLOPs of the previous SoTA method Cross-Stitch). Additionally, our hardware co-design of MoE ViT-tiny achieves energy consumption an order of magnitude lower than Cross-Stitch.

## B.2   Additional Experiments on Different Numbers of Tasks

To evaluate the performance of our model, we further conduct experiments on different levels of MTL difficulties with different numbers of tasks. We compare between STL-B, MTL-B, SoTA work Cross-Stitch [3], MTL-B with ViT-small (M-ViT), and MTL-B with MoE ViT-small ($M^3$ViT); results are shown in Table 2. It can be observed that $M^3$ViT consistently outperforms MTL-B with less computational FLOPs on different numbers of tasks on both NYUD-v2 and PASCAL-Context. Compared to SoTA encoder-focused work Cross-Stitch, although $M^3$ViT performs slightly lower on NYUD-v2 with two tasks, it achieves better performance on all the other settings. In particular, it surpasses Cross-Stitch on NYUD-v2 when the number of tasks increases to four (−0.91% *vs.* −3.26%), which demonstrates the strong capacity of our model on handling more tasks. On PASCAL-Context dataset, introducing MoE ($M^3$ViT) can achieve much better performance than Cross-Stitch. Noticing that $M^3$ViT performs slightly worse on normal estimation and saliency detection tasks, we speculate that it is because these two tasks require a relatively small receptive field to retain a detailed estimation, and Cross-Stitch allows to use limited local information (i.e., small receptive field) when fusing the activations from the different single-task networks. But for other tasks that require larger receptive fields, our model performs significantly better than Cross-Stitch, since our task-dependent

Table 2: Performance on different numbers of tasks

| PASCAL-Context | Backbone | Seg. (mIoU↑) | Norm. (mErr↓) | H. Parts (mIoU↑) | Sal. (mIoU↑) | Edge (odsF)↑ | $\Delta_m$ (%)↑ | FLOPS (G)↓ |
|---|---|---|---|---|---|---|---|---|
| STL-B | ResNet-18 | 66.2 | **13.9** | 59.9 | 66.3 | 68.8 | 0.00 | 167 |
| MTL-B | ResNet-18 | 60.8 | 14.5 | – | – | – | −6.23 | 167 |
| Cross-Stitch [3] | ResNet-18 | 65.4 | 14.2 | – | – | – | −1.68 | 647 |
| M-ViT | MoE ViT-small | 65.3 | 15.6 | – | – | – | −6.79 | **83** |
| $M^3$ViT | MoE ViT-small | **72.7** | 14.4 | – | – | – | **+3.11** | 84 |
| MTL-B | ResNet-18 | 63.8 | 14.9 | 58.6 | 65.1 | 69.2 | −2.86 | 167 |
| Cross-Stitch [3] | ResNet-18 | 66.1 | **13.9** | 60.6 | **66.8** | 69.9 | +0.60 | 647 |
| M-ViT | MoE ViT-small | 70.7 | 15.5 | 58.7 | 64.9 | 68.8 | −1.76 | **83** |
| $M^3$ViT | MoE ViT-small | **72.8** | 14.5 | **62.1** | 66.3 | **71.7** | +2.71 | 84 |

| NYUD-v2 | Backbone | Seg. (mIoU↑) | Depth (rmse)↓ | Norm. (mErr↓) | Edge (odsF)↑ | – | $\Delta_m$ (%)↑ | FLOPS (G)↓ |
|---|---|---|---|---|---|---|---|---|
| STL-B | ResNet-50 | 43.9 | 0.585 | **19.8** | 68.4 | – | **0.00** | 192 |
| MTL-B | ResNet-50 | 44.4 | 0.587 | – | – | – | +0.41 | 192 |
| Cross-Stitch [3] | ResNet-50 | 44.2 | **0.570** | – | – | – | **+1.61** | 310 |
| M-ViT | MoE ViT-small | 40.9 | 0.631 | – | – | – | −6.27 | **100** |
| $M^3$ViT | MoE ViT-small | **45.6** | 0.589 | – | – | – | +1.59 | **100** |
| MTL-B | ResNet-50 | 41.9 | 0.618 | 21.3 | **69.0** | – | −4.22 | 192 |
| Cross-Stitch [3] | ResNet-50 | 42.2 | 0.629 | 20.1 | 68.3 | – | −3.26 | 310 |
| M-ViT | MoE ViT-small | 40.9 | 0.636 | 21.5 | 65.0 | – | −7.28 | **100** |
| $M^3$ViT | MoE ViT-small | **44.8** | **0.612** | 20.1 | 68.6 | – | **−0.91** | **100** |

Table 3: Performance on different numbers of tasks on Taskonomy dataset

| Tasks | Depth | Norm. | Seg. | Edge | Occ. | Reshad. | Key2d. | Curvature | Autoenc. | Average |
|---|---|---|---|---|---|---|---|---|---|---|
| 3 tasks | 3.33% | 0.44% | 7.74% | – | – | – | – | – | – | 3.84% |
| 6 tasks | 4.68% | 2.58% | 10.36% | 0.80% | 3.28% | 8.20% | – | – | – | 4.98% |
| 9 tasks | 5.41% | 1.58% | 7.67% | 0.34% | 4.34% | 5.06% | 7.83% | 0.26% | 15.01% | 5.28% |

MoE design helps effectively avoid different tasks' training conflict. Meanwhile, $M^3$ViT consumes much less computational power than previous methods.

Furthermore, we conduct experiments by choosing tasks from the large-scale Taskonomy dataset [8]. Like our main manuscript, we use MTL-ViT-small as the baseline model and MTL-MoE-ViT-small for our model. We increase the number of tasks from three to nine and perform detailed evaluations. Following the same data pre-processing and evaluation method [9], we report the relative performance improvement from M³ViT over the baseline MTL-ViT. As shown in the Table 3, M³ViT demonstrates even stronger superiority as the number of tasks increases.

## B.3 Comparisons with Decoder-focused Methods

Decoder-focused architectures typically require initial predictions or intermediate features of all the tasks, both in training and inference, to improve the predictions. However, activating all tasks in inference violates our motivation: sparsely activating the network to achieve efficient MTL inference. Moreover, those models consume a large number of FLOPs [10], which makes them difficult to deploy onto real-world edge devices with resource and latency constraints. This is because they need higher parallelism factors, more resources, or clever tricks to hit the desired latency requirement, which is out of scope of the discussion of this paper.

Ignoring the previously mentioned efficiency and memory bottleneck, we conduct comparisons between our $M^3$ViT-base model and decoder-focused work PAD-Net [11], which have similar FLOPs (PAD-Net: 212 GFLOPs vs. Ours: 191 GFLOPs). Our MoE ViT-base model achieves higher performance than PAD-Net on both the PASCAL Context dataset (Ours: +4.0% vs. PAD-Net: -4.41%) and the NYUD-V2 dataset (Ours: +8.32% vs. PAD-Net: +7.43%).

## C   Latency Breakdown of M³ViT

Our FPGA implementation of M³ViT using ViT-small takes 84.538 ms for inference on the NYUD-v2 dataset, which is split between patch embedding, ViT layers, and MoE layers as shown in the breakdown in Figure 1. As shown in this figure, the time required to compute all experts in the MoE layers (18.567 ms) is nearly equal to the time required to compute the fully-connected layers within the ViT layers (18.447 ms). This affirms that our hardware computation reordering mechanism is able to maintain memory efficiency with near-zero impact on latency.

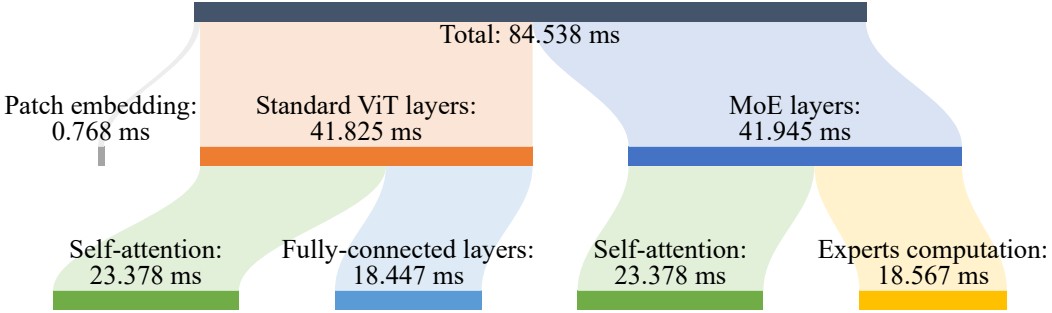

Figure 1: **A breakdown of the FPGA inference latency on the NYUD-v2 dataset.** The total latency can be split into the patch embedding step, the six standard ViT layers, and the six MoE layers in the backbone. The ViT and MoE layers can further be divided into self-attention, which is identical for both types of layers, and either the ViT fully-connected MLPs or the MoE experts computation.