# OpenReview forum: "M³ViT: Mixture-of-Experts Vision Transformer for Efficient Multi-task Learning with Model-Accelerator Co-design"
_NeurIPS.cc/2022/Conference — NeurIPS 2022 Accept_

### Official Review · Reviewer_SvCo · 2022-06-20

**Rating:** 7
**Confidence:** 2
**Soundness:** 3 good
**Presentation:** 4 excellent
**Contribution:** 3 good

**Summary:**

The paper describes an efficient multi-task neural network for vision applications that can be deployed on resource-constrained devices. The model consists of a mixture of vision transformer experts that are sparsely activated by a GELU activation function. The main innovation is that mixture of expert layers are computed expert-by-expert instead of token-by-token, which reduces latencies of loading expert weights.

**Questions:**

* How does the model accuracy and efficiency (Latency, Energy, Memory) depend on the total number of experts N and top experts K?
* Is the softmax in equation 3 needed, given that you are only interested in the top-k activations?
* How are the top-k experts computed at training and inference time? What is the runtime?

**Limitations:**

Yes

**Strengths And Weaknesses:**

## Strengths
* Computing mixture-of-expert layers seems like a simple yet effective why to increase the efficiency of a vision transform mixture of expert architecture
* The paper is well well
* Experiments are solid

## Weaknesses
* Only two datasets (NYUD, PASCAL) are considered in experiments--no real world datasets

---

> ### Author Response · Authors · 2022-08-02
> **Response to Reviewer SvCo**
>
> **[Q1]** Only two datasets (NYUD, PASCAL) are considered in experiments--no real world datasets
> **[A1]** Following the previous MTL paper [1], which conducts a thorough survey on the multi-task dense prediction field, we validate our proposed framework and compare it against other state-of-the-art methods on NYUD-v2 and PASCAL-Context. Both of them are real-world MTL datasets. Additionally, we have conducted more experiments by choosing tasks from the large-scale Taskonomy dataset [2]. Like our main manuscript, we use ViT-small as the baseline model and MoE-ViT-small for our model. We increase the number of tasks from three to nine and perform detailed evaluations.  Following the same data pre-processing and evaluation method as [3], we report the **relative performance improvement** from M³ViT over the baseline ViT. As shown in the table below, M³ViT demonstrates even stronger superiority as the number of tasks increases.
>
> | DeiT-S | Depth zbuffer  | Normal  | Segment semantic  | Edge  | Occlusion  | Reshading  | Keypoints2d | Principal curvature | Auto encoder | Average |
> | :------------: | :----: | :----: | :----: | :----: | :----: | :----: | :----: | :----: | :----: | :----: |
> |3tasks| 3.33% | 0.44% | 7.74% |  |  |  |  |  |  | 3.84%  |
> |6tasks| 4.68% | 2.58% | 10.36% | 0.80% | 3.28% | 8.20% |  |  |  | 4.98% |
> |9tasks| 5.41% | 1.58% | 7.67% | 0.34% | 4.34% | 5.06% | 7.83% | 0.26% | 15.01%  | 5.28% |
>
>
> **[Q2]** How does the model accuracy and efficiency (Latency, Energy, Memory) depend on the total number of experts N and top experts K?
> **[A2]** (1) For a fixed number of tasks, we can fix the total number of expert candidates N and increase per-task expert selection K to encourage feature reuse. In this way, performance can smoothly increase but will quickly saturate, and the training memory and testing energy/latency will also increase.
> (2) A model for a larger number of tasks is likely to benefit from a larger total number of experts N, as more experts can bring larger model capacity. However, if we fix per-task expert count K, the training efficiency and testing energy/latency will not change. This shows the significance of our modularized design in scaling up the number of tasks.
> (3) Based on the above observations, we choose N=16 and K=4, which empirically balance accuracy and efficiency nicely.
>
> **[Q3]** Is the softmax in equation 3 needed, given that you are only interested in the top-k activations?
> **[A3]** Yes, this design has been proven useful in several classic sparse MoE papers [4-6]. We follow their original Top-K implementations, which all introduce a softmax activation function in the output of the gating network. By scaling the logits into a multinomial probability distribution, they can be used to normalize the outputs. If you are referring to the fact that removing softmax would not change the output indices of the Top-K experts, we kindly highlight that we not only use the output indices of the Top-K experts, but also their expert selection scores for representation computations (Equation 2 in the main manuscript).
>
> **[Q4]** How are the top-k experts computed at training and inference time? What is the runtime?
> **[A4.1]** We replace the dense feed-forward network in the ViT block with sparsely activated MoE experts. Each token will be passed into the task-dependent gating network, to select the subset from all expert candidates, in both the training and inference stages.
> **[A4.2]** For an image with a resolution 640x480, the frame rate of our model is 36.0/s on a NVIDIA Quadro RTX 6000 GPU, and on an Xilinx ZCU104 FPGA the frame rate is 11.8/s.
>
> [1] Multi-Task Learning for Dense Prediction Tasks: A Survey
> [2] Taskonomy: Disentangling task transfer learning
> [3] Which Tasks Should Be Learned Together in Multi-task Learning
> [4] OUTRAGEOUSLY LARGE NEURAL NETWORKS: THE SPARSELY-GATED MIXTURE-OF-EXPERTS LAYER
> [5] Gshard: Scaling giant models with conditional computation and automatic sharding
> [6] Switch transformers: Scaling to trillion parameter models with simple and efficient sparsity

---

### Official Review · Reviewer_DMLe · 2022-07-03

**Rating:** 7
**Confidence:** 4
**Soundness:** 3 good
**Presentation:** 4 excellent
**Contribution:** 4 excellent

**Summary:**

This paper studies a novel setting of MTL: training with multiple tasks hoping them to boost each other, and inference with a single task each time. The setting is practically convincing as most MTL systems switch between tasks instead of executing all tasks at the same time. The authors presented a model-accelerator co-design framework, involving both algorithm and hardware innovations.

**Questions:**

Please see the weakness part.

**Limitations:**

No, the current version claimed it has no negative social impacts.

**Strengths And Weaknesses:**

Strong points:
1. The overall problem and idea are novel and interesting. Most MTL papers assume all tasks to be trained and tested in the same bundle. This paper allows for more flexibility.
2. The introduction MoE is well motivated from two aspects: efficiency for single-task inference (sparse activation); and avoiding MTL training conflicts (grouping similar modules). Using MoE is hence natural in this new setting, and to my best knowledge, is novel for the MTL field. The authors also designed a task-dependent gating network for this new purpose.
3. The authors meanwhile present interesting hardware innovations. To my best knowledge, not too many hardware works were done in the MTL domain. This work proposed a novel computation reordering mechanism tailored for memory-constrained MTL and MoE, which allows scaling up to any number of experts and also achieves zero-overhead switching between tasks. It sounds very promising and might be of independent research interest.
4. Experiments are solid, involving customized FPGA design on real hardware, and evaluating two major MTL datasets (NYUD-v2, PASCAL-Context). The proposed co-design can save memory and energy costs by up to one order of magnitude, with no worse accuracies than baselines. Besides, more tasks seem to save more (PASCAL versus NYUD), which shows great scalability to larger numbers of tasks.
5. The abstract and intro were written elegantly to give a proper big picture. The authors also did a good job in making figures. As another plus, codes are provided in the supplementary: I did not have time to carefully check but from quick reading, the code quality looks fine.


Weak points:
1. My main concern is that the MTL baselines used are all somehow weak. Even though the problem setting is new, and I know not many peer algorithms did exactly the same job, there are still some. For example, the authors should check “Task Adaptive Parameter Sharing for Multi-Task Learning” (CVPR 2022). That paper has important similarity to this submission, in learning modularized models for MTL while allowing for single-task inference. Although it does not use dynamic MoE and has no hardware co-design, I believe some level of comparison and discussion is necessary. There might be more in literature.
2. One other baseline-related issue is that the authors only compare with encoder-focused MTL models, and claim that is because decoder-focused ones are in general heavier (but also, more performant). I think the authors should at least compare with or discuss some decoder-focused methods and demonstrate the trade-offs here, rather than completely ignoring. For example, is it because decoder-focused MTL models cannot fit into FPGA, really?
3. Why jumping to a ViT backbone? Looking at Table 1, the ViT baseline is already stronger than most prior works built on ResNet 18 (as expected). Can you compare with other MTL works with the same ViT backbone, or can you replace ViT with Res18 in your co-design framework? That would help disentangle your own contributions.
4. If some MTL applications do require to activate all or most tasks all the time, will your framework still be advantageous? For example, what if always executing 2, 3, or 5 tasks simultaneously on PASCAL? I understand this will change your problem target, but I suggest the authors to include some discussions and clarity for future readers.

---

> ### Author Response · Authors · 2022-08-02
> **Response to Reviewer DMLe**
>
> **[Q1]** My main concern is that the MTL baselines used are all somehow weak. I believe some level of comparison and discussion with “Task Adaptive Parameter Sharing for Multi-Task Learning” is necessary.
> **[A1]** (1) The ViT-small model used in TAPS [1] is bigger than our adopted DeiT-S (Ours: 4.5GFLOPs  vs. TAPS: 9.8GFLOPs). Moreover, [1] was pre-trained on ImageNet-21k while ours was pre-trained on ImageNet-1k. (We have communicated and confirmed this with the authors.) Due to the limited time window of the rebuttal, we don’t have enough time to reimplement their code and pre-train a new model on ImageNet-21k. As their code is not available, we will reimplement their method and provide a fair comparison in our final version.
> (2) Nevertheless, we attempt a comparison with TAPS by conducting experiments on the benchmark: Flowers [2], Cars [3], Sketch [4], CUB [5] and WikiArt [6]. Following the comparisons in Table 3 of TAPS, we compare our method (MoE-DeiT-S) with the fine-tuned model (DeiT-S w/ fine-tuning). We find that our method surpasses the fine-tuned model on most datasets, which validates the effectiveness of our method. However, TAPS only demonstrates comparable results to the fine-tuned ViT-S.
> |DeiT-S| Flowers  | WikiArt  | Sketch  | Cars  | CUB |
> |:----:|:----:|:----:|:----:|:----:|:----:|
> |Fine-tuning| **96.1** | 77.5 | 76.2 | 86.1 | 84.8  |
> |MoE-Deit-S| 95.7 | **79.5** | **79.7** | **86.5** | **85.6** |
>
> **[Q2]** The authors should compare with or discuss some decoder-focused methods and demonstrate the trade-offs.
> **[A2]** (1) These decoder-focused architectures typically require initial predictions or intermediate features of all the tasks, both in training and inference, to improve the predictions. However, activating all tasks in inference violates our motivation: sparsely activating the network to achieve efficient MTL inference. Moreover, those models consume a large number of FLOPs [7], which makes them difficult to deploy onto real-world edge devices with resource and latency constraints. This is because they need higher parallelism factors, more resources, or clever tricks to hit the desired latency requirement, which is out of scope of the discussion of this paper.
> (2) Ignoring the previously mentioned efficiency and memory bottleneck, we conduct comparisons between our M³ViT-base model and decoder-focused work PAD-Net [8], which have similar FLOPs (PAD-Net: 212 GFLOPs vs. Ours: 191 GFLOPs). Our MoE ViT-base model achieves higher performance than PAD-Net on both the PASCAL-Context dataset (Ours: +4.0% vs. PAD-Net: -4.41%) and the NYUD-V2 dataset (Ours: +8.32% vs. PAD-Net: +7.43%).
>
> **[Q3]** Why jumping to a ViT backbone? Looking at Table 1, the ViT baseline is already stronger than most prior works built on ResNet 18. Can you compare with other MTL works with the same ViT backbone, or can you replace ViT with Res18 in your co-design framework?
> **[A3]** We adopt ViT backbones because they are the latest performant deep models, and have achieved impressive performance on various computer vision tasks [9-11]. Although ViT-small-MTL (row 10) achieves better performance than ResNet-MTL (row 2) in Table 1 (ViT-small: -1.77% vs. ResNet-18: -2.86%),  adopting our task-dependent MoE design to ViT **further boosts the performance by a large margin** (Ours: +2.71% vs. ViT-small: -1.77%).
> Apart from that, we will replace ViT with ResNet in our co-design framework and report the results in our later version. Due to the limited time window of the rebuttal, we don’t have enough time to pre-train the MoE-ResNet model.
>
> **[Q4]** If some MTL applications do require to activate all or most tasks all the time, will your framework still be advantageous? I understand this will change your problem target, but I suggest the authors to include some discussions and clarity for future readers.
> **[A4]** Our co-design framework is based on single-task execution. To achieve the goal of multi-task inference, we can design the gating network conditioned on multi-label encodings, to activate model paths for multiple tasks. We will provide a detailed discussion in the future work section.
>
> [1] Task Adaptive Parameter Sharing for Multi-Task Learning.
> [2] Automated flower classification over a large number of classes.
> [3] 3d object representations for fine-grained categorization.
> [4] How do humans sketch objects?
> [5] The caltech-ucsd birds-200-2011 dataset.
> [6] Large-scale classification of fine-art paintings: Learning the right metric on the right feature.
> [7] Multi-Task Learning for Dense Prediction Tasks: A Survey.
> [8] Pad-net: Multitasks guided prediction-and-distillation network for simultaneous depth estimation and scene parsing.
> [9] Vision transformers for dense prediction.
> [10] Segformer: Simple and efficient design for semantic segmentation with transformers.
> [11] Pyramid vision transformer: A versatile backbone for dense prediction without convolutions

---

> > ### Comment · Reviewer_DMLe · 2022-08-10
> > **After response**
> >
> > Thanks for your response. The rebuttal has well addressed my questions. I support this paper for its novelty and solid experiments, and I will keep my original score.

---

### Official Review · Reviewer_V1Gi · 2022-07-09

**Rating:** 4
**Confidence:** 4
**Soundness:** 2 fair
**Presentation:** 2 fair
**Contribution:** 2 fair

**Summary:**

The paper presents a multi-task MoE and model-accelerator co-design. At the algorithm level, the method adopts a task-level mixture-of-expert approach that can effectively reduce training and inference cost. At the model-accelerator level, the paper proposes a computation reordering scheme that is tailored for memory-constrained MTL and achieves little switching overheads. The paper observes significant inference FLOPs reduction and memory requirement reduction, compared to a FPGA baseline.

**Questions:**

1. In table 1, did you implement all the related work similarly on FPGAs? How did you generate the energy numbers? If all the baselines are also implemented in FPGA, then the quality of models are largely determined by implementation not necessarily by the method itself.
2. Two out of six tasks are worse than cross-stitch, any reasons?
3. Why make the total FLOPs only half of the baseline models (Table 1)? A fair comparison would be with similar FLOPs and compare the accuracy.

**Limitations:**

Yes.

**Strengths And Weaknesses:**

Strengths:
- The paper has a very thorough and complete related work and reference section.
- Multi-task MoE is an important field that generates numerous important works. This paper seems to be a right direction addressing important problems. Co-designing a software/hardware systems can be the right problem to solve.
- The proposed reordering reduces memory footprint at small overheads.

Weaknesses:
- The paper is rephrasing many contributions of conditional computation/mixture-of-experts and multi-task MoE. Those contributions are not unique contributions from this paper. For example, token-MoE has been prevalent for a long time and more recently, task-MoE has been demonstrated useful in many tasks including machine translations (e.g. Machine translation from Google). [1]

- The proposed reordering scheme is a system-level contribution which might not best suited for NeurIPS. Even though, there is a co-design claim, the reviewer feels that task-level MoE and and hardware execution reordering should be decoupled. The same reordering scheme can be applied to other types of networks equally. In order to prove co-design is effective, some co-optimization should be applied which can require some joint search space on the MoE model and reordering scheme. The paper should show that co-designing these two parts can be useful and is better than separately optimizing each of them.

- The paper listed challenges round "handle many different tasks" and "switch between tasks" and how different tasks can interfere each other when training on a shared model backbone. An example given in the paper is the autonomous driving setting where potentially hundreds of tasks are running on the shared model backbone. However, the paper does not prove it's own point by evaluating only on a small set of tasks.

[1] "Beyond Distillation: Task-level Mixture-of-Experts for Efficient Inference", https://aclanthology.org/2021.findings-emnlp.304.pdf

---

> ### Author Response · Authors · 2022-08-02
> **Response to Reviewer V1Gi**
>
> **[Q1]** The paper is rephrasing many contributions of conditional computation/mixture-of-experts and multi-task MoE. Those contributions are not unique contributions from this paper.
> **[A1]** Indeed some prior works have used MoE for MTL, but our novelty stands out clearly in several ways, as reiterated below:
> (1) Tailored to efficient on-device MTL, we are the first to explore the novel setting of **multi-task training, single-task inference, and swiftly switching between tasks**. This setting is practically convincing [1-4] and can be uniquely enabled by MoE. No prior work utilizing MoE has ever exploited this setting.
> (2) With our task-dependent MoE and software-hardware co-design, we enable realistic efficient on-device MTL models and demonstrate that MoE for MTL can achieve **real-world memory and energy benefits**. No prior MoE for MTL work, to our best knowledge, has ever accomplished these.
> (3) The introduction of MoE is well-motivated from two aspects (recognized by Reviewer DMLe): resolving cross-task training conflicts and sparsely activating for single-task inference. We respectfully point out that we mainly claim the introduction of MoE as a unified tool for these two purposes (line 90).
>
> **[Q2]** The proposed reordering scheme is a system-level contribution which might not be best suited for NeurIPS. The reviewer feels that task-level MoE and hardware execution reordering should be decoupled.
> **[A2.1]** We respectfully point out that each year, there are dozens of accepted papers in NeurIPS that propose various hardware co-designs and put their hardware-system-level contributions as one of their main claims. Besides, software-hardware co-design for deep neural networks is one of the subfields of “Deep Learning,”  listed in the NeurIPS author guidelines [5]. Below [6-13] is a non-exhaustive list from NeurIPS in recent years only, and the list can go way longer.
> **[A2.2]** Moreover, as ML and FPGA experts, we are very confident on the necessity of our co-design as our hardware innovation is indeed strongly and uniquely tailored for MoE. We reiterate that, if all tokens choose any K experts out of the N candidates, it either requires extreme on-chip memory, or incurs severe delays under a cache-based design (Section 3.2). This is precisely our motivation for the proposed hardware design, which enables zero-overhead switching between tasks and scales to any number of experts. That further lays the foundation for our efficient single-task inference in MTL. Overall, the co-design is tightly integrated that we see no reason to decouple, making it a well noted and well-justified holistic “co-design”. We note that all other reviewers unanimously appreciate this point, quoted as: “co-design for MoE is a timely attempt” (Reviewer YbW8), “hardware design is tailored for memory-constrained MTL and MoE” (Reviewer DMLe), and more.
>
> **[Q3]** The paper does not prove its point of "handle many different tasks" by evaluating only on a small set of tasks.
> **[A3]** (1) In Table 2 of the supplement, we experimentally show that, when we increase the number of tasks, our method consistently demonstrates an increase in the improvement over the baseline method (MTL-B: −2.86% vs. Ours: +2.71% on PASCAL-Context, MTL-B: −4.22% vs. Ours: −0.91% on NYUD-v2). This validates our claim that our method is more effective when handling more tasks.
> (2) To further validate this point, we conduct new experiments by choosing tasks from the large-scale Taskonomy dataset [14]. Like our main manuscript, we use ViT-small as the baseline model and MoE-ViT-small for our model. We increase the number of tasks from three to nine and perform detailed evaluations. Following the same data pre-processing and evaluation method [15], we report the **relative performance improvement** from M³ViT over the baseline ViT. As shown in the table below, M³ViT demonstrates even stronger superiority as the number of tasks increases. We will be happy to integrate those new results into our final draft.
>
> | DeiT-S | Depth zbuffer  | Normal  | Segment semantic  | Edge  | Occlusion  | Reshading  | Keypoints2d | Principal curvature | Auto encoder | Average |
> | :------------: | :----: | :----: | :----: | :----: | :----: | :----: | :----: | :----: | :----: | :----: |
> |3tasks| 3.33% | 0.44% | 7.74% |  |  |  |  |  |  | 3.84%  |
> |6tasks| 4.68% | 2.58% | 10.36% | 0.80% | 3.28% | 8.20% |  |  |  | 4.98% |
> |9tasks| 5.41% | 1.58% | 7.67% | 0.34% | 4.34% | 5.06% | 7.83% | 0.26% | 15.01%  | 5.28% |

---

> > ### Author Response · Authors · 2022-08-02
> > **Response to Reviewer V1Gi**
> >
> > **[Q4]** In table 1, did you implement all the related work similarly on FPGAs? How did you generate the energy numbers? If all the baselines are also implemented in FPGA, then the quality of models are largely determined by implementation not necessarily by the method itself.
> > **[A4.1]** The metrics in Table 1 of the main manuscript are based on standard PyTorch implementations on GPU for all rows, except the “M³ViT  (+ MoE + co-design)”, i.e., the last row. All rows except the last first provide a fair comparison of those algorithms on GPU, showing that our algorithm “M²ViT (+MoE)” outperform all prior work at smaller FLOPs. Then, comparing the second-to-last row and the last row demonstrates the strong energy efficiency gains of our hardware co-design on FPGA.
> > **[A4.2]** Energy metrics are computed as the product of the power consumption of the target device (GPU for all rows except the last; FPGA for the last row) in Watts and inference latency in seconds.
> > **[A4.3]** Furthermore, we provide cross-platform and cross-model performance improvement breakdowns in Table 4 for a fairer comparison. Specifically, compared to a naive implementation on FPGA, our hardware co-design using computation re-ordering decreases the energy consumption of MoE ViT on FPGA from 6.375 W·s to 0.690 W·s on the PASCAL-Context dataset.
> >
> > **[Q5]** Two out of six tasks are worse than cross-stitch, any reasons?
> > **[A5]** The reasons are two-fold:
> > (1) Cross-stitch [16] network has a more complex network design (Cross-stitch: 647 GFLOPs vs. Ours: 84 GFLOPs).
> > (2) Both normal estimation and saliency detection tasks require a relatively small receptive field to retain a detailed estimation, and [16] is only allowed to use limited local information (i.e., small receptive field) when fusing the activations from the different single-task networks [17].  But for other tasks that require larger receptive fields, our model performs significantly better than Cross-stitch, since our task-dependent MoE design helps effectively avoid different tasks’ training conflict. We will add those discussions into our final draft.
> >
> > **[Q6]** Why make the total FLOPs only half of the baseline models (Table 1)? A fair comparison would be with similar FLOPs and compare the accuracy.
> > **[A6]** (1) We use half the FLOPs because our backbone is MoE-ViT-small, while the baseline model uses ResNet18 as backbone (rows 1, 2). We adopt ViT as it is the latest performant deep model, which has achieved impressive performance on various computer vision tasks [18-20].
> > (2) Please note for a fair comparison, we also report a baseline model based on ViT-small (Table 1, row 10). Although using a ViT-small backbone helps to achieve better performance than the ResNet baseline (ViT-small: -1.77% vs. ResNet-18: -2.86%), introducing our task-dependent MoE design **further boosts the performance by a large margin** (Ours: +2.71% vs. ViT-small: -1.77%).
> > (3) Meanwhile, in Table 1 of the supplement, we also report comparisons between baseline models (row 2, 7) and our MoE-ViT-base model (row 5, 10), with similar FLOPs (MTL-B: 167G vs. Ours: 161G, MTL-B: 192G vs. Ours: 191G). Our model boosts the accuracy by a large margin (Ours: +4.0% vs. MTL-B: -2.86% on PASCAL-Context dataset and Ours: +8.32% vs. MTL-B: +0.41% on NYUD-v2 dataset).
> >
> > [1] Fast drivable areas estimation with multi-task learning for real-time autonomous driving assistant
> > [2] Indoor semantic segmentation for robot navigating on mobile
> > [3] Task Inference and Distributed Task Management in the Centibots Robotic System
> > [4] Evolutionary swarm robotics: genetic diversity, task-allocation and task-switching
> > [5] https://nips.cc/Conferences/2022/CallForPapers
> > [6] Mest: Accurate and fast memory-economic sparse training framework on the edge. (NeurIPS2021)
> > [7] Learning Semantic Representations to Verify Hardware Designs. (NeurIPS2021)
> > [8] Hardware-adaptive efficient latency prediction for nas via meta-learning. (NeurIPS2021)
> > [9] Shiftaddnet: A hardware-inspired deep network. (NeurIPS2020)
> > [10] Learning-in-the-loop optimization: End-to-end control and co-design of soft robots through learned deep latent representations. (NeurIPS2019).
> > [11] Constrained deep neural network architecture search for IoT devices accounting for hardware calibration. (NeurIPS2019).
> > [12] Towards hardware-aware tractable learning of probabilistic models. (NeurIPS2019).
> > [13] Hardware conditioned policies for multi-robot transfer learning. (NeurIPS2018).
> > [14] Taskonomy: Disentangling task transfer learning
> > [15] Which Tasks Should Be Learned Together in Multi-task Learning
> > [16] Cross-stitch Networks for Multi-task Learning
> > [17] Multi-Task Learning for Dense Prediction Tasks: A Survey
> > [18] Segformer: Simple and efficient design for semantic segmentation with transformers
> > [19] Pyramid vision transformer: A versatile backbone for dense prediction without convolutions
> > [20] Vision transformers for dense prediction.

---

> > ### Comment · Reviewer_V1Gi · 2022-08-05
> > **Thanks for the response.**
> >
> > Thanks the authors for the timely and detailed rebuttal. After reading other reviews and the rebuttal, I have a few additional questions:
> > 1) How the proposed double buffered computation strategy is novel as compared to the Ping-pong buffer? In your rebuttal to Review YbW8, you agree that "the double-buffering strategy is also known as ping-pong buffering. We will add a reference to Xilinx, “Specifying Arrays as Ping-Pong Buffers or FIFOs” [5] to make it more clear." As I pointed out in my previous review, multi-task MoE does not seem to be very novel, as compared to Google's task-MoE for translation, your response further makes me doubt the novelty of this work. Please help clarify. If neither task-moe nor double buffering stragegy is not new, or there is no joint design space, I would not agree with the author about the definition of co-design.
> >
> > 2) Resonating with Review V1Gi, why do you choose a ViT backbone? The reviewer feels that baselines around ViT targeting edge devices or FPGA's with very limited on-chip bandwidth can be naturally very bad, compared to SoTA models built around ConvNets. The reviewer would like to see evaluations built around SoTA ConvNet models. To my understanding, ResNet 18 is not a strong baseline targeting mobile devices or FPGA and ViT beats ResNet 18 is not surprising. Instead, you could compare with MobileNet-V3 or EfficientNetV1 or V2 or ShuffleNet.

---

> > > ### Author Response · Authors · 2022-08-08
> > > **Response to Reviewer V1Gi**
> > >
> > > **[Q1]** How the proposed double buffered computation strategy is novel as compared to the Ping-pong buffer? In your rebuttal to Review YbW8, you agree that "the double-buffering strategy is also known as ping-pong buffering. We will add a reference to Xilinx, “Specifying Arrays as Ping-Pong Buffers or FIFOs” [5] to make it more clear." As I pointed out in my previous review, multi-task MoE does not seem to be very novel, as compared to Google's task-MoE for translation, your response further makes me doubt the novelty of this work. Please help clarify.
> > > **[A1.1]** Sorry for your confusion, but you seem to have misread our hardware innovation as well as rebuttal context. We clarify our hardware novelties as follows:
> > > (1) To adapt MoE ViT on hardware with acceptable latency and power, we first propose an effective per-expert queue design to enable expert-by-expert computation rather than token-by-token. Our design uses O(1) on-chip memory so that it can scale to any K and N (line 228-229) and eliminate task-switch and frame-switch overhead in our MTL system (line 229-232). No prior work, to our best knowledge, has ever achieved these.
> > > The designing philosophy is clear (lines 205-215): computing each token normally either requires all N experts stored on-chip, incurring **extreme on-chip memory usage** that scales with O(N), or requires an on-chip cache of experts that causes **severe memory delays due to frequent cache misses**. Thus our model is **infeasible to implement in hardware** until we introduce our hardware co-design.
> > > (2) While double-buffering/ping-pong buffering itself is a well-established technique, compared with previous frameworks, our co-design introduces double-buffering in a totally different new way. To be more specific, the novelty of our design is that we propose a new computation flow that makes double-buffering effective where it otherwise would not be.
> > > Double-buffering is only effective at hiding latency when computation latency dominates the memory access latency. Fulfilling this prerequisite is not trivial. For instance, in the cache-based hardware design (lines 210-215), severe memory delays are incurred by continuously reloading the cache (line 215). As a result, memory access latency will dominate and double-buffering cannot alleviate this latency bottleneck.
> > > Our expert-by-expert reordering unifies each expert’s memory accesses (lines 217-220), creating a novel scheme for MTL MoE ViT execution where computation latency dominates memory latency. Thus we propose to adopt double-buffering to hide memory latency almost completely (lines 227-228), where it otherwise would be ineffective. As this technique is unique to our work, it has never been exploited by any prior work utilizing double-buffering.
> > > **[A1.2]** As mixture-of-experts provides the scaffold upon which our computation reordering strategy is built, we reiterate that we propose a novel co-design framework for efficient on-device MTL, where software and hardware contributions are strongly and uniquely **TIED** and cannot be decoupled. Our hardware optimizations are predicated on an MoE ViT algorithm design where each token requires any K of the N total experts, and they solve the hardware challenges of extreme on-chip memory usage and extreme latency associated with such an algorithm design. Furthermore, they ensure zero task-switch overhead, necessary for our MTL system, which has fast task switching as a primary goal.

---

> > > > ### Author Response · Authors · 2022-08-08
> > > > **Response to Reviewer V1Gi**
> > > >
> > > > **[Q2]** Resonating with Review V1Gi, why do you choose a ViT backbone? The reviewer feels that baselines around ViT targeting edge devices or FPGA's with very limited on-chip bandwidth can be naturally very bad, compared to SoTA models built around ConvNets. The reviewer would like to see evaluations built around SoTA ConvNet models. To my understanding, ResNet 18 is not a strong baseline targeting mobile devices or FPGA and ViT beats ResNet 18 is not surprising. Instead, you could compare with MobileNet-V3 or EfficientNetV1 or V2 or ShuffleNet.
> > > > **[A2.1]** We adopt ViT backbones because they are the latest performant deep models, and have achieved impressive performance on various computer vision tasks [9-11]. While ViT-small-MTL (row 10) achieves better performance than ResNet-MTL (row 2) in Table 1 (ViT-small: -1.77% vs. ResNet-18: -2.86%), adopting our task-dependent MoE design to ViT further boosts the performance by a large margin (Ours: +2.71% vs. ViT-small: -1.77%).
> > > > **[A2.2]** We respectfully draw attention to the numerous existing works which explore transformer development on edge devices and model acceleration. As the latest performant deep models, targeting ViT on edge devices or FPGA is well motivated and practically mature now - in both academia and industry. Below [1-7] is a non-exhaustive list of only the most recent literature.
> > > > **[A2.3]** (1) We compare against ResNet-18 because several state-of-the-art MTL dense prediction frameworks [8-11] are all developed based on ResNet-18.
> > > > (2) We also change our backbone to MobileNet-V3-large [12] and test on the PASCAL-Context dataset. The backbone is pre-trained on ImageNet-1k and we load the pretrained weights from [13]. We can see from the table below, MobileNet-v3 performs even worse than the ResNet-18 baseline. We speculate that it is because MobileNet-v3 is a more compact model, during training, the gradient conflicts between different tasks are even more severe. We also notice that the previous MTL on dense prediction frameworks tend to adopt ResNet backbones[14], rather than highly compact ones (e.g., MobileNet). Meanwhile, our proposed M³ViT achieves much higher MTL accuracy while requiring fewer inference FLOPs, coupled with the novel software-hardware co-design. This demonstrates that our model efficiently balances feature reuse with compact model capacity and avoids conflict between different tasks; both are enabled by our MoE design. We will release the code of ResNet-MTL, MobileNet-MTL, and M³ViT.
> > > > | Model | Seg. ↑ | Norm. ↓| H. Parts ↑ | Sal. ↑ | Edge ↑ | $\Delta(m)$ ↑ |FLOPS(G)|
> > > > | :----: | :----: | :----: | :----: | :----: | :----: |:----: | :----: |
> > > > |ResNet18-MTL | 63.8 | 14.9 | 58.6 | 65.1 | 69.2  |-2.86 | 167  |
> > > > |MobileNet-MTL|56.7 |18.8 |48.8 | 58.5 |63.1 |-17.6 |157 |
> > > > |M³ViT-small| 72.8 | 14.5 | 62.1 | 66.3 | 71.7 | +2.71 | 83 |
> > > >
> > > > [1] Qi, Panjie, et al. “Accommodating Transformer onto FPGA: coupling the balanced model compression and FPGA-implementation optimization.” In Proceedings of the 2021 on Great Lakes Symposium on VLSI, 2021.
> > > > [2] Liu, Zejian, et al., “Hardware acceleration of fully quantized BERT for efficient natural language processing.” Design, Automation & Test in Europe Conference & Exhibition (DATE). IEEE, 2021.
> > > > [3] Peng, Hongwu, et al. “Accelerating Transformer-based deep learning models on FPGAs using column balanced block pruning.” International Symposium on Quality Electronic Design (ISQED), IEEE, 2021.
> > > > [4] Li, Bingbing, et al., “FTRANS: energy-efficient acceleration of Transformers using FPGA.” In Proceedings of the ACM/IEEE International Symposium on Low Power Electronics and Design, 2020
> > > > [5] Sun, Mengshu, et al., “VAQF: fully automatic software-hardware co-design framework for low-bit Vision Transformer”, arXiv 2022
> > > > [6] Kong, Zhenglun, et al., “SPViT: enabling faster Vision Transformers via soft token pruning”, arXiv 2021
> > > > [7] He, Jiaao, et al.  ''Fastmoe: A fast mixture-of-expert training system." arXiv 2021
> > > > [8] Cross-stitch networks for multi-task learning
> > > > [9] Latent multitask architecture learning
> > > > [10] NDDR-CNN: Layerwise feature fusing in multi-task cnns by neural discriminative dimensionality reduction
> > > > [11] End-to-end multi-task learning with attention
> > > > [12] Searching for MobileNetV3
> > > > [13] https://github.com/d-li14/mobilenetv3.pytorch
> > > > [14] Multi-Task Learning for Dense Prediction Tasks: A Survey

---

> > > ### Author Response · Authors · 2022-08-08
> > > **Response to Reviewer V1Gi**
> > >
> > > Dear Reviewer V1Gi:
> > >
> > > Since the author-reviewer discussion period will end by tomorrow, we will appreciate if you could check our response to your review comments soon.
> > >
> > > If our response resolves your concerns, we kindly ask you to consider raising the rating of our work.
> > >
> > > Thank you very much for your time and efforts

---

> ### Author Response · Authors · 2022-08-05
> **Response to Reviewer V1Gi**
>
> Dear Reviewer V1Gi:
>
> Since the author-reviewer discussion period has started for a few days, we will appreciate if you could check our response to your review comments soon.
>
> If you have further questions and comments, we can still reply before the author-reviewer discussion period ends. If our response resolves your concerns, we kindly ask you to consider raising the rating of our work.
>
> Thank you very much for your time and efforts.

---

### Official Review · Reviewer_YbW8 · 2022-07-11

**Rating:** 6
**Confidence:** 2
**Soundness:** 3 good
**Presentation:** 3 good
**Contribution:** 3 good

**Summary:**

In this paper, the authors propose to solve the gradient conflict in training and dense activation in inference in multi-task learning using a co-designed mixture-of-expert model. In algorithm level, the authors propose to replace ViT layer with mixture of expert layer and sparsely activate them during training to alleviate the gradient conflict issue, while during inference, the author propose a novel computation reordering scheme to better support the sparse activation for different tasks. Experiments on GPU and FPGA show the proposed co-design achieves better performance than existing methods.

**Questions:**

- How are different experts shared between different tasks? Will the scaling in task number requires more experts to maintain a satisfied performance?
- What is the training cost of the proposed model compared with baseline methods? Does it need extra training to learn the expert-task mapping and fully train the large number of experts?
- What is the performance improvement breakdown for the proposed hardware design?

**Limitations:**

The authors have adequately addressed the limitations and potential negative societal impact of their work.

**Strengths And Weaknesses:**

Strengths:
- The co-design for Mixture-of-Expert to enable multi-task learning is a timely attempt.
- The achieved results is promising in both accuracy and efficiency.

Weaknesses:
- Using mixture-of-expert to deal with multi-task learning is not the novel idea, but the co-design attempt is interesting to me.
- Is the mentioned double buffered computation strategy similar to Ping-pong buffer? If so it would be better to add the reference.

---

> ### Author Response · Authors · 2022-08-02
> **Response to Reviewer YbW8**
>
> **[Q1]** Using mixture-of-expert to deal with multi-task learning is not a novel idea, but the co-design attempt is interesting to me.
> **[A1]** We agree that some prior works use MoE for MTL, but we respectfully point out that:
> (1) Tailored to efficient on-device MTL, we are the first to explore the novel setting of using multi-task training with single-task inference and swiftly switching between tasks, which is practically convincing [1-4] and can be uniquely enabled by MoE. No prior work utilizing MoE has ever exploited this setting.
> (2) By proposing the task-conditional MoE MTL ViT and the hardware innovations, we enable realistic efficient on-device MTL models and demonstrate that incorporating MoE into MTL can achieve significant memory and energy benefits in real-world systems. No prior work, to our best knowledge, has ever accomplished these.
>
> **[Q2]** Is the mentioned double buffered computation strategy similar to Ping-pong buffer? If so, it would be better to add the reference.
> **[A2]** Yes, the double-buffering strategy is also known as ping-pong buffering. We will add a reference to Xilinx, “Specifying Arrays as Ping-Pong Buffers or FIFOs” [5] to make it more clear.
>
> **[Q3]** How are different experts shared between different tasks? Will the scaling in task number require more experts to maintain a satisfied performance?
> **[A3.1]** Each of the different tasks will select its own top-K experts from the total of N expert candidates via a task-dependent gating network. Expert sharing across tasks is learned automatically through training.
> **[A3.2]** (1) Yes. A model for a larger number of tasks is likely to benefit from a larger total number of experts N, as more experts can bring a larger model capacity. (2) However, we do not need to scale up the per-task expert count K when more tasks are involved. (3) The training efficiency and inference resource cost are dependent only on K and per-expert size. Our results on both NYUD-v2 and PASCAL-Context (Table 1 in appendix) also validate that our proposed co-design model can scale up nicely with more tasks.
>
> **[Q4]** What is the training cost of the proposed model compared with baseline methods? Does it need extra training to learn the expert-task mapping and fully train the large number of experts?
> **[A4.1]** M³ViT has a very similar number of training FLOPs to the ViT baseline, as (1) we only activate a small portion of experts for each image token, and (2) the computational cost incurred by the added gating network, which is a one-layer MLP network per ViT block, is negligible (~0.05 GFLOPs for image resolution 640x480).
> **[A4.2]** We do not need extra training for the expert-task mapping. In our experiments, we keep the number of training epochs of MoE-ViT the same as that of the ViT baseline.
>
> **[Q5]** What is the performance improvement breakdown for the proposed hardware design?
> **[A5]** Our proposal aims to improve the latency of the “experts computation”. For example, vanilla “experts computation” takes 684.586ms in a cache-based FPGA implementation on the NYUD dataset, and is reduced to 18.567ms with our hardware co-design—a 97.3% improvement. Please refer to Table 4 in the main manuscript for more details on the overall performance improvement with our hardware co-design, as well as Figure 2 in the supplement for a latency breakdown.
>
> [1] Fast drivable areas estimation with multi-task learning for real-time autonomous driving assistant
> [2] Indoor semantic segmentation for robot navigating on mobile
> [3] Task Inference and Distributed Task Management in the Centibots Robotic System
> [4] Evolutionary swarm robotics: genetic diversity, task-allocation and task-switching
> [5] https://docs.xilinx.com/r/en-US/ug1399-vitis-hls/Specifying-Arrays-as-Ping-Pong-Buffers-or-FIFOs

---

### Author Response · Authors · 2022-08-02
**Summary of Author's Response**

We thank all reviewers for their insightful and constructive suggestions. We are glad that reviewers found
(1) The introduction of MoE for multi-task learning (MTL) is well motivated (Reviewer DMLe);
(2) The model-accelerator co-design for efficient on-device MTL is an attempt in the right direction (Reviewer YbW8, V1Gi, DMLe, SvCo), and it provides promising performance and might be of independent research interest (Reviewer DMLe);
(3) Experiments are solid (Reviewer DMLe, SvCo) and the proposed framework achieves good accuracy and/or efficiency (Reviewer YbW8, V1Gi).

We have addressed all the questions that the reviewers posed with additional experimental results. We will carefully modify our main manuscript later, following those suggestions.

---

### Meta-Review · Area_Chair_4iKk · 2022-08-26

**Recommendation:** Accept
**Confidence:** Certain

**Metareview:**

This paper presents a model-accelerator co-design framework to enable on-device Multi-task Learning (MTL). At the model level, customized mixture-of-expert (MOE) layers are introduced for MTL, which alleviate gradient conflict at training time and improve the efficiency at inference time via sparse activation. At the accelerator level, the paper proposes computation reordering which allows zero-overhead switching between tasks. The algorithm is verified the on popular multi-task datasets, and the accelerator is implemented on commercial FPGAs, demonstrating improved efficiency.

The paper is very well written, the details on the algorithm and hardware implementation are clearly explained. The author chose a particular setting of MTL, then design the model and tailor the parameters to enable efficient on-device MTL. The work is complete, covering from algorithm design to hardware implementation with sufficient innovations.

Reviewers have raised concerns such as
1). Evaluation on small datasets. During rebuttal period, the authors provide more experimental results from the large-scale Taskonomy dataset.
2). Overclaiming. For example, double buffering is a well-known technique for dataflow optimization. The technique itself is by no means novel. However, I think using it to solve a practical problem still has value.

Overall, it is a solid paper and is recommended for acceptance.


**Award:**

No

---

### Decision · Program_Chairs · 2022-09-14

Accept